# Hybrid Polyetheretherketone (PEEK)–Acrylic Resin Prostheses and the All-on-4 Concept: A Full-Arch Implant-Supported Fixed Solution with 3 Years of Follow-Up

**DOI:** 10.3390/jcm9072187

**Published:** 2020-07-10

**Authors:** Miguel de Araújo Nobre, Carlos Moura Guedes, Ricardo Almeida, António Silva, Nuno Sereno

**Affiliations:** 1MALO CLINIC, Research and Development Department, Av. Combatentes, 43, 11, 1600-042 Lisbon, Portugal; 2MALO CLINIC, Prosthodontics Department, Av. Combatentes, 43, 10, 1600-042 Lisbon, Portugal; cguedes@maloclinics.com (C.M.G.); ralmeida@maloclinics.com (R.A.); 3MALO CLINIC, Ceramics, Av. Combatentes, 43, 11, 1600-042 Lisbon, Portugal; amsilva@maloceramics.com; 4Invibio Biomaterial Solutions & JUVORA, Global Technology Center, Hillhouse International, Thornton, Cleveleys FY5 4QD, UK; nsereno@invibio.com

**Keywords:** dental implants, immediate dental implant loading, polyetheretherketone, PEEK, prostheses and implants

## Abstract

Background: The aim of this three-year prospective study was to examine the outcome of a solution for full-arch rehabilitation through a fixed implant-supported hybrid prosthesis (polyetheretherketone (PEEK)-acrylic resin) used in conjunction with the All-on-4 concept. Methods: Thirty-seven patients (29 females, 8 males), with an age range of 38 to 78 years (average: 59.8 years) were rehabilitated with 49 full-arch implant-supported prostheses (12 maxillary rehabilitations, 13 mandibular rehabilitations and 12 bimaxillary rehabilitations). The primary outcome measure was prosthetic survival. Secondary outcome measures were marginal bone loss, plaque and bleeding scores, veneer adhesion issues, biological complications, mechanical complications, and the patients’ subjective evaluation. Results: There were two patients (maxillary rehabilitations) lost to follow-up, while one patient withdrew (maxillary rehabilitation). One patient with bimaxillary rehabilitation fractured the mandibular PEEK framework, rendering a 98% prosthetic survival rate. Implant survival was 100%. Average (standard deviation) marginal bone loss at 3-years was 0.40 mm (0.73 mm). Veneer adhesion was the only technical complication (*n* = 8 patients), resolved for all patients. Nine patients (*n* = 11 prostheses) experienced mechanical complications (all resolved): fracture of acrylic resin crowns (*n* = 3 patients), prosthetic and abutment screw loosening (*n* = 4 patients and 3 patients, respectively), abutment wearing (*n* = 1 patient). One patient experienced a biological complication (peri-implant pathology), resolved through non-surgical therapy. A 90% satisfaction rate was registered for the patients’ subjective evaluation. Conclusions: Based on the results, the three-year outcome suggests the proposed rehabilitation solution as a legitimate treatment option, providing a potential shock-absorbing alternative that could benefit the implant biological outcome.

## 1. Introduction

Polyetheretherketone (PEEK) is a high-performance thermoplastic polymer with high strength-to-weight ratio and corrosion resistance that makes it suitable as a selectable material to replace metal [1].

Considering its original development (Victrex plc, Lancashire, UK) [2], the fabrication process [3] results in a number of properties including chemical stability, biostability, biocompatibility, creep and wear resistance, and superior mechanical behavior. These properties allow compatibility with medical diagnostic imaging [3,4], extending its use from industrial [3] applications to those in the fields of medicine [5,6,7,8,9] and dentistry [6,10,11,12,13,14,15].

In the field of dentistry, the possibility of computer-aided design/computer-aided manufacturing (CAD/CAM) together with its biocompatibility and shock absorbing features [10,16,17] has enabled the increased use of PEEK. These uses were extended to the fabrication of a considerable number of materials including healing caps, abutments, removable prostheses, crowns, fixed or partial full-arch dentures, and dental implants [10,11,12,13,14,15,18,19,20,21,22,23]. Despite the scarce in vivo publications, PEEK was demonstrated to be a valid treatment option compared to titanium in the rehabilitation of severely atrophic maxillary alveolar ridges both through patient-specific sub-periosteal implants [24] and as a customized mesh for bone augmentation [25].

From the psychological point of view, immediate function protocols have proven to be a benefit for the patient [26] due to the immediate restoration of mastication, phonetics, and esthetics [27]. The All-on-4 concept (Nobel Biocare, Zurich, Switzerland) involves inserting four implants for immediate function full-arch rehabilitation of edentulous jaws [28,29]. This is achieved by strategically placing four implants: two implants placed posterior with up to 45 degrees of angulation and two anterior implants in an axial orientation, providing stability in the presence [6,7] or limited absence of high primary stability (≥30 N·cm) [30]. The long-term outcome of full-arch rehabilitation via the All-on-4 concept was recently evaluated: A high cumulative survival rate (CSR) was registered in the maxilla (94.7% implant CSR with up to 13 years of follow-up) [31] and mandible (93% implant CSR with up to 18 years of follow-up) [32]. In addition, the breakdown of the All-on-4 concept survival can be evaluated considering the implant CSR of 97.6% to 100% under two years [33,34,35,36], 96–99% in three–five years [29,37,38,39], and 95.4–100% in 5–10 years [28,40,41,42]. The All-on-4 concept was further validated as a treatment option by two systematic reviews [43,44].

A previous study reported on the one-year evaluation of a hybrid solution (PEEK infrastructure with acrylic resin artificial gingiva and acrylic resin teeth) for fixed full-arch rehabilitations with the All-on-4 concept. The report registered promising short term outcomes considering the implants’ survival and marginal bone loss, as well as high patient satisfaction [20] The aim of this three-year prospective study was to examine the prosthetic and implant outcomes of full-arch rehabilitations through a fixed implant-supported hybrid prosthesis (PEEK-acrylic resin) used in conjunction with the All-on-4 concept.

## 2. Materials and Methods

This prospective cohort clinical study was conducted between May 2015 and October 2019 in a private practice (Lisbon, Portugal). Patients with need of full-arch implant-supported rehabilitations experienced surgical and prosthetic intervention between May 2015 and October 2016. The study was approved by an independent ethical committee (Ethical Committee for Health, authorization No. 008/2013), (registered ClinicalTrials.gov ID NCT04446078) and all patients provided written informed consent. The study included 37 full-arch edentulous patients (29 females and 8 males) with an age range of 38 to 78 years (average (standard deviation): 59.8 years (10.6 years)); 12 patients with bimaxillary rehabilitations, 12 patients with maxillary rehabilitations and 13 patients with mandibular rehabilitations (total of 49 edentulous arches). The dataset was made available in an open access repository (https://osf.io/hmyux/).

### 2.1. Inclusion and Exclusion Criteria

At the treatment planning phase, inclusion and exclusion criteria were identified. Patients were included provided they had been rehabilitated with implant-supported fixed prostheses through the All-on-4 concept (Nobel Biocare); while patients in active chemotherapy or radiotherapy, presenting insufficient bone volume, or unable to provide written informed consent were excluded.

### 2.2. Surgical and Prosthetic Protocols

The interventions (both surgical and prosthetic protocols for All-on-4 rehabilitation) have been depicted in previous publications [20,28,29,39,45,46]. In brief, the implants (Nobelspeedy^TM^, Nobel Biocare AB, Zurich, Switzerland) were inserted following standard procedures [45] except for the use of under-preparation to guarantee a final torque of over 32 N·cm before the final implant seating. The implant length ranged between 10 and 18 mm. Four implants were inserted using distal tilting for the posterior implants (30 to 45 degrees) and two anterior implants were placed in an axial position providing support for an immediate implant-supported fixed prosthesis of high-density acrylic resin with a minimum of 10 teeth.

### 2.3. Manufacture and CAD/CAM Guidelines

As pointed out in the previous short-term report [20], the CAD/CAM guidelines relating to the cross-sectional material dimensions when designing and manufacturing the PEEK framework were: an “I” shape framework design with a minimum anterior buccal-lingual width of 4 mm, a minimum occlusal-cervical height of 5 mm, an increased width in the areas of the titanium sleeve to allow 6 mm of minimum buccal-lingual width, and a minimum of 1–2 mm of acrylic resin with the objective of increasing adhesion considering the previously mentioned dimensions. At the CAM phase, a titanium sleeve was introduced within the PEEK infrastructure’s coronal aspect (encircled by PEEK and acrylic resin) with no special retention mechanism. This was done to prevent the titanium prosthetic screws strangulating the PEEK upon torque tightening.

### 2.4. Definitive Prosthetic Protocol

The definitive prosthetic protocol and CAD/CAM guidelines were described in detail in a previously published short-term outcome report [20]. In brief, the definitive screw-retained implant-supported prostheses provided to each patients’ arch was a hybrid polymer-acrylic resin featuring a PEEK infrastructure (Juvora, Ltd., Lancashire, United Kingdom), titanium sleeves (patent No. WO2019/008368 A1) [47] as an aid to the prosthesis-abutment interface, pink acrylic resin gingiva (PalaXpress Ultra, Heraeus Kulzer GmbH, Hanau, Germany), and 12 acrylic resin teeth (Premium and Mondial crowns, Heraeus Kulzer GmbH) [34]. In nine patients (*n* = 9 prostheses), the cantilever length was more than one unit. A mutually protected occlusion plan was chosen. Figure 1 illustrates a clinical case representative of the present study.

After the connection of the full-arch definitive prostheses, the patients were evaluated every six months (clinically) and at one and three years for function (clinically and radiographically).

### 2.5. Outcome Measures

We evaluated prosthetic survival as primary outcome measure (considering the necessity of replacing the prosthesis).

We evaluated implant survival, marginal bone loss, plaque scores, bleeding scores, problems during manufacturing, complications (both biological and mechanical), and the evaluation performed by the patient as secondary outcome measures.

We evaluated implant survival considering the implants’ function, censoring as a failure the first implant to fail in a given patient [3]. This implied that, considering all patients had four implants supporting the prosthesis, the failure of one of the implants was marked as a failure for the patient irrespective if the remaining three implants remained in function.

The authors examined marginal bone loss through periapical radiographs employing a radiographic holder (super-bite; Hawe Neos, Bioggio, Switzerland), adjusted for the digital film’s orthognathic position. The radiographs were evaluated by an outcome assessor through software for image analysis (rayMage, version 2.3, MyRay, Imola, Italy). Marginal bone level was defined as the distance between the implant’s platform and the most apical bone-implant contact, while the measurement difference between baseline (connection of the definitive prosthesis) and the three years evaluation was classified as marginal bone loss (MBL). We calibrated the measurements using the distance between implant threads and considered average values between mesial and distal sites [20].

Plaque levels and bleeding levels were evaluated according to the modified plaque index (mPLI) and modified bleeding index, respectively, using an ordinal scale [48].

Problems during manufacturing were evaluated considering veneer adhesion and framework integrity.

We classified adverse soft tissue reaction, suppuration, abscess, fistulae and peri-implant pathology (the presence of marginal bone loss and peri-implant pockets of more than 4 mm, with or without the concurrent presence of bleeding on probing or suppuration) as biological complications.

We classified prosthesis, abutments or prosthetic screw fracture or loosening as mechanical complications.

The evaluations performed by the patient were registered on a visual analogue scale ranging between 0–10 (poor to excellent) and comprised the aspects: “in-mouth comfort”, specified as the patients’ overall evaluation according to expectations when submitting the prosthesis to function, and “overall chewing feeling”, associated with the specific feeling during food mastication [20].

### 2.6. Statistical Analysis

The variables prosthetic survival and implant survival were analyzed using the prosthesis and implant as the unit of analysis, respectively. We estimated the cumulative survival rate (CSR) through life tables using the actuarial method. We calculated the mean (with 95% confidence intervals) and standard deviation for the variables: age, MBL and the evaluations performed by the patient; for modified plaque index (mPLI) and modified bleeding index (mBI) the median was estimated; while for problems during manufacturing, biological complications and mechanical complications, we estimated frequencies. We examined the correlation between mPLI and mBI through Spearman’s coefficient of correlation considering *p* < 0.05 as significant. The software Statistical Package for the Social Sciences (IBM SPSS, New York, NY, USA) version 17 was used to analyze the data.

## 3. Results

### 3.1. Sample

In this study, a total of 49 full-arch restorations with an average cantilever length of 6.79 mm (standard deviation: 5.66 mm; range: 0–16.5 mm) per prosthesis were connected in 37 patients. There were two patients (5.4%) with maxillary prostheses (4.1%) lost to follow-up (throughout the first six months) as they were inaccessible, while one patient (2.6%) with a full-arch maxillary prosthesis (2%) withdrew from the study.

### 3.2. Primary Outcome Measure

One male patient (55 years old, heavy bruxer) with bimaxillary rehabilitation, fractured the lower arch PEEK framework prosthesis (with consequent need of replacement), rendering a 98% prosthetic CSR (Table 1).

### 3.3. Secondary Outcome Measures

One-hundred-and-ninety-six implants were placed in order to rehabilitate a total of 49 edentulous arches. All implants remained in function, rendering a three-year 100% CSR.

The mean (standard deviation) MBL at 1- and 3-years was 0.37 mm (0.58 mm) and 0.40 mm (0.73 mm), respectively. Considering the MBL evaluation per arch at 1- and 3- years, the maxillary implants registered 0.33 mm (0.52 mm) and 0.38 mm (0.77 mm), while the mandibular implants registered 0.40 mm (0.63 mm) and 0.42 (0.70 mm), respectively (Table 2).

The scores for mPLI and mBI were characterized at six months by median scores of 1 (plaque only visible after performing the test) and 1 (isolated bleeding spot visible). On the mid-term outcome, the mPLI and mBI scores remained unaltered between 1- and 3-years of follow-up, with a median score of 2 (corresponding to visible plaque) and 1 (isolated bleeding spot visible), respectively (Figure 2 and Figure 3 depict the mid-term outcome).

A weak positive correlation according to the Spearman’s correlation coefficient was registered between mPLI and mBI throughout the study period at 6-months, 1- and 3-years (Figure 4). (*R* = 0.408; *p* = 0.017).

Veneer adhesion problems (acrylic resin avulsion from PEEK infrastructure) occurred in 10 prostheses (20.4%) at the prosthesis level (Table 3).

The incidence of biological complications was registered at 29 months in one implant (0.5%) that was localized in position #35, consisting of peri-implant pathology: the implant exhibited a peri-implant pocket of 5 mm, concurrent bleeding on probing, and a MBL of 1.6 mm. The complication was successfully treated by non-surgical therapy, comprised of scaling, 0.2% chlorhexidine gel irrigation (Periokin gel, Kin, Barcelona, Spain), and oral hygiene instructions. No further incidences of biological complications occurred. Nine patients (24.3%) and 11 prostheses (22.5%) exhibited mechanical complications (Table 4) and were mainly found in patients with bimaxillary rehabilitations having, as opposing dentition, an implant-supported fixed prosthesis.

The subjective evaluation for satisfaction performed by the patient showed similar averages (standard deviation) at 1- and 3-years (Figure 5) of 88% (16%) and 90% (9.5%) for “in-mouth comfort” and 84% (19%) and 90% (13.6%) for “overall chewing feeling” (Figure 5).

## 4. Discussion

The present study reports the three-year outcome of a restoration solution for full-arch edentulism comprised of a fixed hybrid polymer-acrylic prosthesis with CAD/CAM infrastructure, as an alternative to other recent CAD/CAM solutions [49].

The 98% prosthetic CSR was influenced by a mandibular prosthetic failure in a bimaxillary patient presenting bruxism habit that caused a framework fracture, suggestive of occlusal overload. In a previous systematic review, Hsu et al. [50] registered bruxism and occlusal overload to be the primary etiologic factors for biomechanical complications. Moreover, previous investigations on the effect of bruxism and sex on the maximum human occlusal force recorded 978–1000 N peaks in male heavy bruxers [51,52]. In this study, despite PEEK’s 1200 N module of deformation point (referring to the change in size or shape by an applied force) [10], the constant application of forces of this magnitude on the prosthetic materials in daily use had a negative influence on the prosthetic outcome of one patient.

The CAD/CAM guidelines for the framework (described in the Materials and Methods section) were created to compensate for material flexion, allowing a maximum of one cantilever unit and 1–2 mm acrylic resin for increased adhesion. Given the exploratory nature of this study, more than one cantilever unit (>10 mm) was inserted in nine prostheses of nine patients, rendering technical complications (veneer adhesion issues) more common in prosthesis with increased cantilevers, which suggested flexing of the PEEK framework distal cantilever as the potential cause. However, it should be noted that all problems related to the adhesion between PEEK and acrylic resin materials were solved regardless of the cantilever length and without prejudice to implant and prosthetic survival, underlining the importance of the proper bonding primer.

A previous study investigating All-on-4 restorations and considering different implant distributions reported a protective effect for absence of cantilever units (odds ratio = 0.22) and a risk effect for bruxism (odds ratio = 60.95) when examining mechanical complications [46]. In this study, similar findings were registered: bruxism influenced the incidence of mechanical complications as represented by the framework and prosthetic fractures in one patient.

The registered 100% implant CSR of the present study is comparable to recent publications for the All-on-4 concept. Maló et al. (2019) [31,32] reported three-year CSR of 97.8% and 99.1% for the maxilla [31] and mandible [32]. Furthermore, a systematic review evaluating the All-on-4 treatment concept registered a 99.8% implant CSR for All-on-4 restorations with a follow-up of two or more years [44].

The mean MBL at three years of follow-up demonstrated the stability of the rehabilitations over this period, as compared to the difference of 0.03 mm of mean MBL registered in the previous one-year report [20]. Moreover, the MBL recorded in the present study at three years compares favorably with previous studies reporting a range of 1.06–1.52 mm for the maxilla [29,53] and 1.30 mm overall for both maxilla and mandible [41]. In a systematic review [54] comparing marginal bone loss between axial and tilted implant-supported fixed prosthetic reconstructions, a range of 0.91–1.55 mm and 0.72–1.67 mm was registered after three years of follow-up of full-arch restorations for axial and tilted implants, respectively (data extracted from study). The potential explanation of this lower MBL could be associated with PEEK’s characteristics of shock absorption, given its 1200 N cut-off point for plastic deformation (important in load-bearing areas) [10].

The median for mPLI was elevated, corresponding to visible plaque (score 2 of the mPLI scale) and consistent with inadequate oral hygiene habits; while the isolated bleeding spots around the implants recorded as median mBI level was considered mild (score 1 of the mBI scale). The causal association between plaque accumulation and mucositis has been reported by other authors [55,56] who observed an increased severity of mucositis (±1 mm increase in peri-implant probing depths accompanied by inflammation) [55] that suggested a positive linear relationship [56] when patients ceased all self-care efforts for three weeks (with undisturbed plaque accumulation). Additionally, they observed a negative linear relationship between plaque and bleeding scores (decrease in both indices) once the period of optimal plaque control was restored [56], claiming causality between plaque buildup and decreased peri-implant health. The same was not confirmed in the present study given the significant yet weak positive correlation between mPLI and mBI scores. From an epidemiological point of view, analyzing this result in conjunction with the very low incidence of biological complications registered in this study opens the possibility that peri-implant pathology could be triggered by more than one causal mechanism, instead of the disease development being purely driven by a biofilm-mediated infection. Furthermore, the biomechanical component could play a role in the causal mechanism considering the component causal model [57] as previously suggested [58,59].

The study registered a significantly important rate of veneer adhesion problems between PEEK and acrylic resin that the authors attribute to the learning curve and the choice of an inappropriate bonding agent. Veneer adhesion problems related to mechanical and chemical retentions between the acrylic resin and PEEK infrastructure led to the use of a bonding primer with a higher tensile bond strength, supported by the results of previous investigations [11,60], in order to resolve the adhesion issues. Moreover, the cylinder area was reinforced by increasing the amount of exposed PEEK. These modifications were introduced taking into consideration PEEK flexion capacity and were detailed in the previous one-year report [20] Adhesion issues were considered part of the learning curve in the manufacturing process of the PEEK-acrylic resin restorations. Considering the authors’ experience, it is advisable to comply with the following measures (additional to the CAD/CAM guidelines previously described) to prevent veneer adhesion issues: choose a correct bonding agent that enables strong chemical retention; provide a rough finish, vertical threads in the cantilever area and a horizontal thread in the remaining PEEK infrastructure (not a smooth and round finish); and enable an increased amount of exposed PEEK on the cylinder areas (Figure 6).

With respect to the mechanical complications, the 22.5% incidence at prosthetic level was significantly influenced by their occurrence in bimaxillary rehabilitated patients (seven out of nine patients). This trend was reported in a previous investigation comparing bimaxillary and single-arch patients using the same All-on-4 rehabilitation protocol, where a significant difference in mechanical complications was registered in bimaxillary rehabilitated patients during the five-year investigation [61].

The incidence of biological complications was limited to one episode of peri-implant pathology, which was resolved non-surgically. The percentage of peri-implant pathology (0.5%) compares favorably to other studies using the same rehabilitation protocol (All-on-4) in the same period. Maló et al. reported a 3% and 2.1% incidence at implant level during the first three years of follow-up for maxillary [31] and mandibular rehabilitations [32], respectively. The incidence rate was also lower as compared to a previously published systematic review that reported a peri-implant pathology with a five-year estimate of 5.4% at the implant level [62]. Considering the influence of biomechanical factors as risk indicators for peri-implant disease [58], it is important to consider the potential positive influence of the PEEK framework on this lower incidence. This is based on the previously discussed biomechanical properties of the PEEK material [10], together with the present study’s low mBI scores compared to the high mPI scores. Nevertheless, a longer follow-up of at least five years is mandatory to evaluate this outcome with better precision.

The patients’ overall satisfaction with the restorations was reflected by the very high values (90%) scored in both classes, “in-mouth comfort” and “overall chewing feeling”. Moreover, it compares favorably to both the one-year patient evaluation of the same prostheses (88% and 84%, respectively) [20] and previous studies [63] that recorded a 79% satisfaction rate in both the fulfillment of expectations and improvement in chewing ability for patients rehabilitated with fixed mandibular full-arch implant-supported fixed prostheses of metal-acrylic resin.

The present study has limitations, including being a single center study, with a small sample size, a short follow-up and absence of a control group, and therefore the results should be interpreted with caution. The strengths of this study relate to the prospective design and the low rate of dropouts (8%), rendering increased internal validity. Nevertheless, the dropout rate can significantly influence the survival rate, since patients dropping out have an increased probability of negative outcomes; therefore, the prosthetic survival should be interpreted carefully as it could be overestimated. Future research should aim to examine both the five years outcome and the evaluation of a routine group.

## 5. Conclusions

Based on the results and within the study limitations, the three-year outcome suggests the proposed full-arch hybrid PEEK-acrylic resin fixed rehabilitation solution as a legitimate treatment option, albeit pending lengthier evaluation. The study registered a high prosthetic/implant CSR and patient satisfaction, a low MBL, and low rate of biological and mechanical complications. This treatment modality provides a potential shock-absorbing alternative that could benefit the implant biological outcome, but it should be further studied in a longer follow-up.

## 6. Patents

An International patent resulting from the work reported in this manuscript was issued on 10 January 2019: Silva, A.; Legatheaux, J.; de Araújo Nobre, M.; Guedes, C.M.; Almeida, R.; Maló, P.; Sereno, N. Dental prosthesis. International patent no. WO 2019/008368 A1.

## Figures and Tables

**Figure 1 jcm-09-02187-f001:**
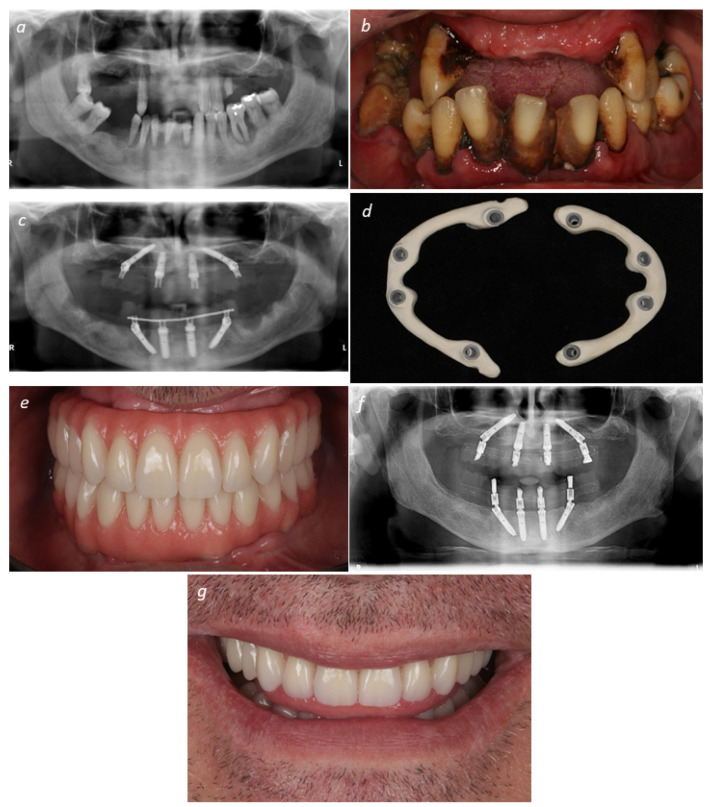
Representative clinical case illustrating a bimaxillary full-arch rehabilitation through a PEEK (Juvora, Ltd.) – acrylic resin (Heraeus Kulzer GmbH) prostheses supported by dental implants by the All-on-4 concept (Nobel Biocare): (**a**) Pre-treatment orthopantomography; (**b**) Pre-treatment frontal view; (**c**) Post-treatment orthopantomography after bimaxillary restoration; (**d**) Polyetheretherketone (PEEK) substructure after computer-assisted manufacture (CAM); (**e**) Final bimaxillary rehabilitation; (**f**) Final post-treatment orthopantomography; (**g**) Post-treatment frontal view of patient smile line.

**Figure 2 jcm-09-02187-f002:**
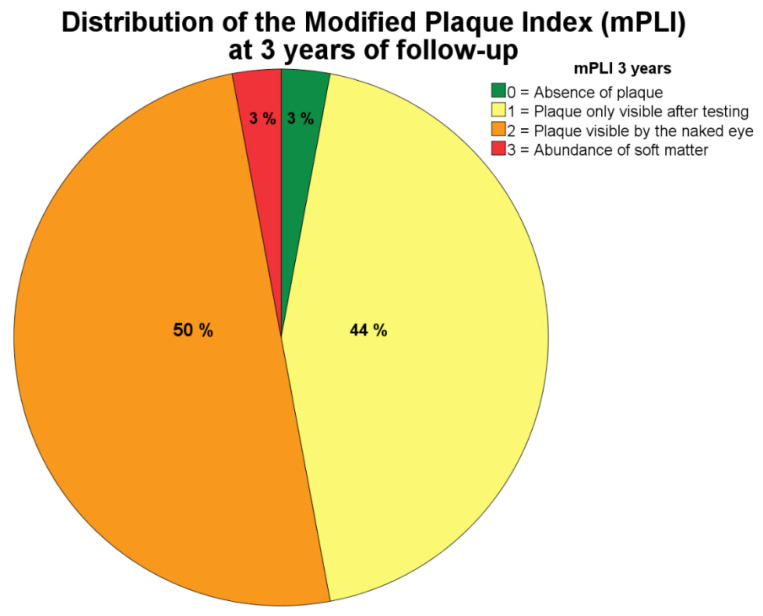
Distribution of modified plaque index (mPLI) at three years of follow-up. Note that majority of patients show level 2 (visible plaque).

**Figure 3 jcm-09-02187-f003:**
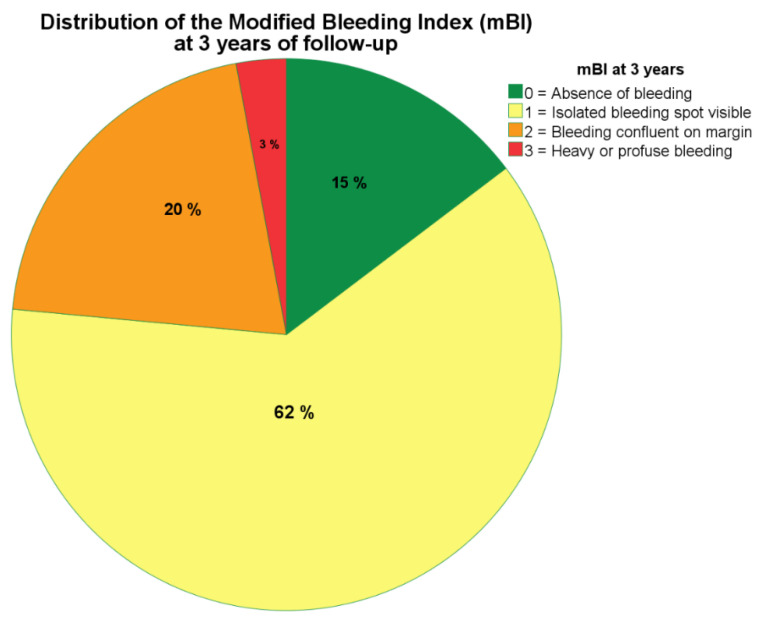
Distribution of the scores for modified bleeding index (mBI) at three years of follow-up. Note that majority of patients show level 1 (isolated bleeding spot visible) on the mBI.)

**Figure 4 jcm-09-02187-f004:**
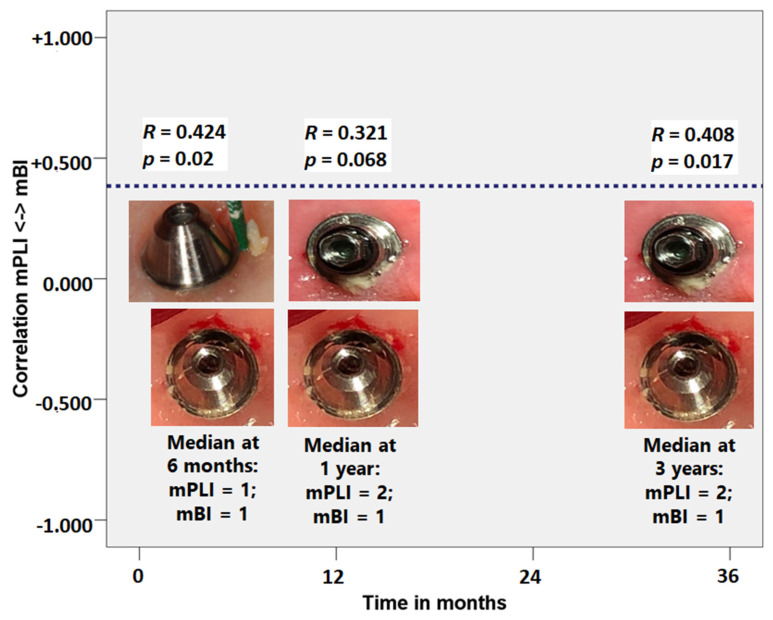
Correlation coefficient between plaque scores (modified plaque index - mPLI) and bleeding scores (modified bleeding index - mBI) at 6 months, 1 year and 3 years of follow-up. Note the correlations were always characterized by weak correlations.

**Figure 5 jcm-09-02187-f005:**
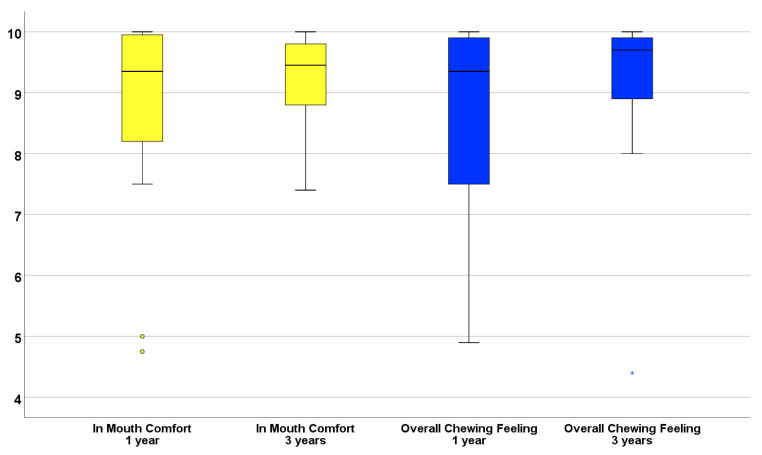
Box-plot illustrating patients’ evaluation while using the polyetheretherketone (PEEK)-acrylic resin prostheses at 1- and 3-years. Note that median satisfaction levels (horizontal black line inside each box) for “In mouth comfort” and “Overall chewing feeling” were registered above 80% (84% and 88%, respectively) at one-year and at 90% (for both evaluations) at three-years.

**Figure 6 jcm-09-02187-f006:**
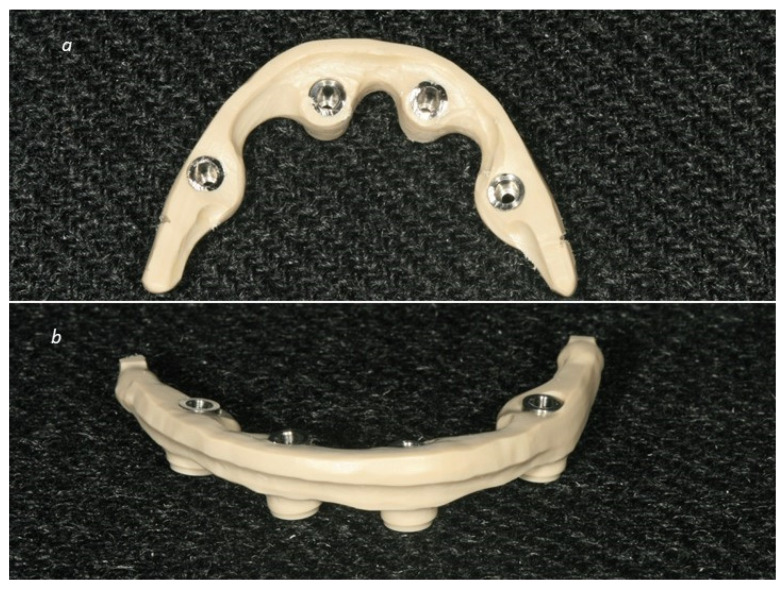
Additional guidelines to prevent veneer adhesion issues. (**a**) Infrastructure inferior view: note the finish, vertical threads on the cantilever area and the enabling of an increased amount of exposed polyetheretherketone (PEEK) on the cylinder areas; (**b**) Infrastructure superior view: note the horizontal thread in the remaining PEEK infrastructure (not smooth and round finish).

**Table 1 jcm-09-02187-t001:** Prosthetic cumulative survival rate (CSR) for hybrid polyetheretherketone (PEEK)—acrylic resin prosthetic restorations.

Time	Total Number of Patients	Prostheses				
		Total Number of Prostheses	Prosthetic Failures	Lost to Follow-Up	Withdrawn	CSR
Prosthesis connection–1 year	37	49	1	2 ^a^	0	98.0
1 year–2 years	35	46	0	0	0	98.0
2 years–3 years	35	46	0	0	1 ^b^	98.0

^a^ Two prostheses in two patients. ^b^ One prosthesis in one patient.

**Table 2 jcm-09-02187-t002:** Average marginal bone loss (95% confidence intervals [CI]) and standard deviation (St. dev.) at 1- and 3-years of follow-up for single arch and bimaxillary rehabilitations.

Site		All Implants		Medial Implants		Distal Implants	
		1 Year	3 Years	1 Year	3 Years	1 Year	3 Years
Maxilla Single arch	Average (mm) [95% CI]:	0.38 [0.21–0.54]	0.50 [0.22–0.77]	0.51 [0.23–0.78]	0.68 [0.41–0.94]	0.24 [0.04–0.45]	0.32 [0.00–0.82]
	St. dev. (mm):	±0.83	±0.81	±0.13	±0.54	±0.10	±1.01
Mandible Single arch	Average (mm) [95% CI]:	0.49 [0.28–0.70]	0.48 [0.26–0.70]	0.57 [0.25–0.90]	0.55 [0.23–0.86]	0.41 [0.11–0.71]	0.41 [0.09–0.74]
	St. dev. (mm):	±0.11	±0.77	±0.16	±0.76	±0.14	±0.79
Bimaxillary (Total)	Average (mm) [95% CI]:	0.29 [0.17–0.41]	0.32 [0.18–0.46]	0.31 [0.15–0.47]	0.26 [0.10–0.42]	0.27 [0.08–0.46]	0.38 [0.14–0.62]
	St. dev. (mm):	±0.61	±0.67	±0.08	±0.53	±0.09	±0.79
Bimaxillary (Maxilla)	Average (mm) [95% CI]:	0.29 [0.10–0.47]	0.29 [0.06–0.51]	0.21 [0.04–0.37]	0.22 [0.00–0.48]	0.38 [0.00–0.77]	0.36 [0.00–0.74]
	St. dev. (mm):	±0.09	±0.73	±0.08	±0.58	±0.18	±0.86
Bimaxillary (Mandible)	Average (mm) [95% CI]:	0.51 [0.16–0.86]	0.35 [0.17–0.54]	0.50 [0.00–1.21]	0.31 [0.09–0.52]	0.53 [0.05–1.00]	0.40 [0.07–0.73]
	St. dev. (mm):	±0.16	±0.61	±0.28	±0.48	±0.19	±0.73

**Table 3 jcm-09-02187-t003:** Veneer adhesion problems between acrylic resin and polyetheretherketone (PEEK) infrastructure and resolution.

Patient	Gender	Follow Up (Months)	Position (FDI)	Type Rehabilitation	Opposing Dentition	Resolution
1	Male	5	#12, #22, #25, #35	Bimaxillary	Implant-supported prosthesis	New prostheses due to fracture of PEEK infrastructure
2	Male	2	#35	Mandibular	Mucosal-retained full-arch prosthesis	To increase flexion resistance, the cylinder areas were left with increased amounts of exposed PEEK; to increase mechanical retention in PEEK infrastructure, a tungsten bur was used; to increase tensile bond strength, the bonding primer was replaced
3	Female	4	#46	Mandibular	Natural teeth and implant-supported prosthesis	
4	Female	10	#45	Mandibular	Mucosal-retained full-arch prosthesis	
5	Female	12	#35	Mandibular	Mucosal-retained full-arch prosthesis	
6	Female	12	#15, #22	Bimaxillary	Implant-supported prosthesis	
7	Female	16	#26	Maxillary	Natural teeth	
8	Female	30	#35	Mandibular	Implant-supported prosthesis	
9	Male	32	#12	Maxillary	Implant-supported prosthesis	

**Table 4 jcm-09-02187-t004:** Incidence of mechanical complications and resolutions during the three years of the study.

Patient	Gender	Opposing Dentition	Cantilever Units (Left/Right) in mm	Follow Up in Months	Acrylic Resin Crown Fracture (Position FDI)	Abutment Wearing (Position FDI)	Abutment Loosening (Position FDI)	Prosthetic Screw Loosening (Position FDI)	Resolution
1	Male	ISP	0/0 (maxilla); 10/10 (mandible)	5	#12,22,#35				1; Patient fractured PEEK infrastructure
2	Male	ISP	13.25/5	16	#32		#42		1
3	Male	ISP	13.25/16.5	22	#41				1
4	Female	NT	10/10	15		#45			2
5	Female	ISP	10/5	16			#45	#42	3
6	Female	ISP	16.5/16.5	16			#42		3
7	Female	ISP	10/10 (maxilla); 0/0 (mandible)	8				#25,#35, #45	3
8	Female	ISP	0/0	4				#15	3
9	Male	NT	11/11	20				#16, #26	3

ISP: Implant supports prosthesis; NT: Natural Teeth. Resolutions: 1—Mending the prostheses and adjusting occlusion; 2—Replacing the abutment and adjusting occlusion; 3—Torque controlled retightening and adjusting occlusion.

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
