# Peer review of "Hybrid Polyetheretherketone (PEEK)–Acrylic Resin Prostheses and the All-on-4 Concept: A Full-Arch Implant-Supported Fixed Solution with 3 Years of Follow-Up"

_jcm, 2020, doi:10.3390/jcm9072187_

Round 1

Reviewer 1 Report

Abstract –

Ok.

Intro –

This is quite vague are some of the paragraphs are 1 sentence or incredibly long 1 sentences.  I would include more information on survival of non-PEEK all-on-4 and PEEK vs. other materials (say, Ti) in other situations besides all-on-4.

Was this trial registered anywhere?

Methods –

Curious why there were substantially more females than males?

We evaluated implant survival considering the implants’ function, censoring as a failure the first implant to fail in a given patient [3]. – I do not follow this sentence.

I would like more information in the text on the implants themselves beyond the citations.

Discussion/results –

The resolution column of Table 3 is confusing.

Figure 4 is a box plot? I do not see it?!

(Paragraph 1, discussion) 1200 N module of deformation point – I do not follow

(Paragraph 2, discussion) Given the exploratory nature of this study – I do not really think this was an exploratory study considering it follows 1 year results?

The discussion is generally quite useful.

Conclusion-

The study also registered a somewhat important rate of veneer adhesion problems between PEEK and acrylic resin that the authors attribute to the learning curve and the choice of an inappropriate bonding agent. – I do not really see a lot of discussion of this point in the manuscript so perhaps add more discussion of it or replace the sentence

Author Response

Abstract –

  1.  

Response: Thank you. 

Intro –

  1. This is quite vague are some of the paragraphs are 1 sentence or incredibly long 1 sentences.  I would include more information on survival of non-PEEK all-on-4 and PEEK vs. other materials (say, Ti) in other situations besides all-on-4.

Response: The authors thank the Reviewer’s indications. The Introduction received a major revision in content and format including all the Reviewer’s suggestions.

Changes: Page 2.

  1. Was this trial registered anywhere?

Response: The authors thank the Reviewer’s query. The study was not registered at its inception but it is registered now. The receipt is attached as a supplementary file. 

Changes: Page 2.

Methods –

  1. Curious why there were substantially more females than males?

Response: The authors thank the Reviewer’s query. There was no particular reason other that the inclusion was performed consecutively when the patients fulfilled the inclusion criteria.

Changes: None.

  1. We evaluated implant survival considering the implants’ function, censoring as a failure the first implant to fail in a given patient [3]. – I do not follow this sentence.

Response: The authors thank the Reviewer’s query. The sentence was clarified, meaning that in a full-arch rehabilitation supported by four implants, even if one of the implants failed and three other implants remained in function and even when the prosthesis could be supported by the remaining implants, the case was still flagged as a failure on implant survival evaluation.

Changes: Page 5. “This implied that considering all patients had 4 implants supporting the prosthesis, the failure of one of the implants was marked as a failure for the patient irrespective if the remaining 3 implants remained in function.”

  1. I would like more information in the text on the implants themselves beyond the citations.

Response: The authors thank the Reviewer’s indication. More information on the implants themselves and insertion protocol was introduced in the text as requested by the Reviewer.

Changes: Page 3, “implant insertion (NobelspeedyTM, Nobel Biocare AB) followed standard procedures [32] with the exception of the use of under-preparation was employed to guarantee a final torque of over 32 N-cm before the final implant seating. The implant length ranged between 10 and 18 mm.”

Discussion/results –

  1. The resolution column of Table 3 is confusing.

Response: The authors thank the Reviewer’s indication. The table format was fixed.

Changes: Page 8, Table 3.

  1. Figure 4 is a box plot? I do not see it?!

Response: The authors thank the Reviewer’s indication. Figure 4 was introduced.

Changes: Page 9, Figure 4.

  1. (Paragraph 1, discussion) 1200 N module of deformation point – I do not follow

Response: The authors thank the Reviewer’s query. The deformation point of a material refers to the force necessary to apply in order to change the size or shape. The sentence was improved for clarity.

Changes: Page 10 “In this study, despite PEEK’s 1200 N module of deformation point (referring to the change in size or shape by an applied force) [10], the constant application of forces of this magnitude on the prosthetic materials in daily use had a negative influence on the prosthetic outcome.”

  1. (Paragraph 2, discussion) Given the exploratory nature of this study – I do not really think this was an exploratory study considering it follows 1 year results?

Response: The authors thank the Reviewer’s indication. The term exploratory study is used in the sense of the process of investigating a problem that has not been studied or thoroughly investigated in the past. And because the current study, despite following up the 1 year results, still uses the same sample from its inception and bears the burden of the choices made in the sampling stage.

Changes: None.

  1. The discussion is generally quite useful.

Response: Thank you. 

Conclusion-

  1. The study also registered a somewhat important rate of veneer adhesion problems between PEEK and acrylic resin that the authors attribute to the learning curve and the choice of an inappropriate bonding agent. – I do not really see a lot of discussion of this point in the manuscript so perhaps add more discussion of it or replace the sentence

Response: The authors thank the Reviewer’s indication. The sentence was moved to the Discussion section also complying with the request of the other Reviewer of inserting more Discussion on the avoidance of this event.

Changes: page 11 and 12.

Reviewer 2 Report

Dear Authors,

I've really appreciated Your manuscript because of its clinical value.

This is a common and useful restoration way, that could be economically, functionally and aesthetically valid.

I would suggest just some minor change before publication.

  • Please in keyword section use Medical Subject Headings (MeSH word)
  • At the end of introduction section please specify the aim of the study
  • Veeners failure is a common event in this type of rehabilitation, please better specify how to avoid it, some figure could be useful.
  • In discussion section please add some recent literature to resin and fixed implant restoration: https://doi.org/10.3390/prosthesis2020006
  • In conclusion section please specify the advantage, and the future perspective of the study

Author Response

  1. Dear Authors,

I've really appreciated Your manuscript because of its clinical value.

This is a common and useful restoration way, that could be economically, functionally and aesthetically valid.

I would suggest just some minor change before publication.

Response: Thank you for your constructive review.

  1. Please in keyword section use Medical Subject Headings (MeSH word)

Response: The authors thank the Reviewers’s indication. MeSH terms only were introduced as requested.

Changes: Page 1: “dental implants; immediate dental implant loading; PEEK”

  1. At the end of introduction section please specify the aim of the study

Response: The authors thank the Reviewer’s indication. The aim of the study was introduced at the end of the introduction as requested.

Changes: Page 2. “The aim of this 3-year prospective study was to examine the prosthetic and implant outcomes of a solution for the full-arch rehabilitation through a fixed implant-supported hybrid prosthesis (polyetheretherketone (PEEK)-acrylic resin) used in conjunction with the All-on-4 concept.”

  1. Veeners failure is a common event in this type of rehabilitation, please better specify how to avoid it, some figure could be useful.

Response: The authors thank the Reviewer’s suggestion. An explanation and figure were introduced as requested, also complying with another request from the other Reviewer.

Changes: Page 11, Figure 5.

  1. In discussion section please add some recent literature to resin and fixed implant restoration: https://doi.org/10.3390/prosthesis2020006

Response: The authors thank the Reviewer’s suggestion. The reference was introduced in the discussion as suggested.

Changes: Page 10, “The present study reports the 3-year outcome of a restoration solution for full-arch edentulism comprised of a fixed hybrid polymer-acrylic prosthesis with CAD/CAM infrastructure, as an alternative to other recent CAD/CAM solutions [Tallarico 2020].”

  1. In conclusion section please specify the advantage, and the future perspective of the study

Response: The authors thank the Reviewer’s indication. The advantage and future perspective of the study were introduced as indicated.

Changes: Page 12, “This treatment modality provides a potential shock-absorbing alternative that could benefit the implant biological outcome but it should be further studied in a longer follow-up.”

Round 2

Reviewer 1 Report

All my have questions have been answered, thanks.